# Efficacy of Artificial Intelligence-Assisted Discrimination of Oral Cancerous Lesions from Normal Mucosa Based on the Oral Mucosal Image: A Systematic Review and Meta-Analysis

**DOI:** 10.3390/cancers14143499

**Published:** 2022-07-19

**Authors:** Ji-Sun Kim, Byung Guk Kim, Se Hwan Hwang

**Affiliations:** 1Department of Otolaryngology-Head and Neck Surgery, Eunpyeong St. Mary’s Hospital, College of Medicine, Catholic University of Korea, Seoul 03312, Korea; skswltjs23@hanmail.net (J.-S.K.); entkbg@gmail.com (B.G.K.); 2Department of Otolaryngology-Head and Neck Surgery, Bucheon St. Mary’s Hospital, College of Medicine, Catholic University of Korea, Bucheon 14647, Korea

**Keywords:** mouth neoplasms, imaging, optical image, precancerous conditions, artificial intelligence, screening

## Abstract

**Simple Summary:**

Early detection of oral cancer is important to increase the survival rate and reduce morbidity. For the past few years, the early detection of oral cancer using artificial intelligence (AI) technology based on autofluorescence imaging, photographic imaging, and optical coherence tomography imaging has been an important research area. In this study, diagnostic values including sensitivity and specificity data were comprehensively confirmed in various studies that performed AI analysis of images. The diagnostic sensitivity of AI-assisted screening was 0.92. In subgroup analysis, there was no statistically significant difference in the diagnostic rate according to each image tool. AI shows good diagnostic performance with high sensitivity for oral cancer. Image analysis using AI is expected to be used as a clinical tool for early detection and evaluation of treatment efficacy for oral cancer.

**Abstract:**

The accuracy of artificial intelligence (AI)-assisted discrimination of oral cancerous lesions from normal mucosa based on mucosal images was evaluated. Two authors independently reviewed the database until June 2022. Oral mucosal disorder, as recorded by photographic images, autofluorescence, and optical coherence tomography (OCT), was compared with the reference results by histology findings. True-positive, true-negative, false-positive, and false-negative data were extracted. Seven studies were included for discriminating oral cancerous lesions from normal mucosa. The diagnostic odds ratio (DOR) of AI-assisted screening was 121.66 (95% confidence interval [CI], 29.60; 500.05). Twelve studies were included for discriminating all oral precancerous lesions from normal mucosa. The DOR of screening was 63.02 (95% CI, 40.32; 98.49). Subgroup analysis showed that OCT was more diagnostically accurate (324.33 vs. 66.81 and 27.63) and more negatively predictive (0.94 vs. 0.93 and 0.84) than photographic images and autofluorescence on the screening for all oral precancerous lesions from normal mucosa. Automated detection of oral cancerous lesions by AI would be a rapid, non-invasive diagnostic tool that could provide immediate results on the diagnostic work-up of oral cancer. This method has the potential to be used as a clinical tool for the early diagnosis of pathological lesions.

## 1. Introduction

Oral cancer accounts for 4% of all malignancies and is the most common type of head and neck cancer [1]. The diagnosis of oral cancer is often delayed, resulting in a poor prognosis. It has been reported that early diagnosis increases the 5-year survival rate to 83%, but if a diagnosis is delayed and metastasis occurs, the survival rate drops to less than 30% [2]. Therefore, there is an urgent need for early and accurate detection of oral lesions and for distinguishing precancerous and cancerous tissues from normal tissues.

The conventional screening method for oral cancer is visual examination and palpation of the oral cavity. However, the accuracy of this method is highly dependent on the subjective judgment of the clinician. Diagnostic methods such as toluidine blue staining, autofluorescence, optical coherence tomography (OCT), and photographic imaging were useful as adjunctive methods for oral cancer screening [3,4,5,6].

Over the past decade, studies have increasingly showed that artificial intelligence (AI) technology is consistent with or even superior to human experts in identifying abnormal lesions in additional images of various organs [7,8,9,10,11]. These results give us hope for the potential of AI in the screening of oral cancer. However, large-scale statistical approaches to diagnostic power for using oral imaging with AI are lacking. Therefore, in this study, the sensitivity and specificity were analyzed through meta-analysis to evaluate the accuracy of detecting oral precancerous and cancerous lesions in AI-assisted oral mucosa images. We also performed subgroup analysis to determine whether accuracy differs between imaging tools.

## 2. Materials and Methods

### 2.1. Literature Search

Searches were performed in six databases: PubMed, Embase, Web of Science, SCOPUS, Cochrane Central Register of Controlled Trials, and Google Scholar. The search terms were: “artificial intelligence”, “photo”, “optical image”, “dysplasia”, “oral precancer”, “oral cancer”, and “oral carcinoma”. The search period was set to June 2022, and data written in English were reviewed. Two independent reviewers reviewed all abstracts and titles of candidate studies. Among studies diagnosing oral cancer using images, studies that did not deal with AI were excluded.

### 2.2. Selection Criteria

The inclusion criteria were: (1) use of AI; (2) prospective or retrospective study protocol; (3) comparison of AI-assisted screening of oral mucosal lesions with the reference test (histology); and (4) sensitivity and specificity analyses. The exclusion criteria were: (1) case report format; (2) review article format; (3) diagnosis of other tumors (laryngeal cancer or nasal cavity tumors); and (4) lack of diagnostic AI data. The search strategy is summarized in Figure 1.

### 2.3. Data Extraction and Risk of Bias Assessment

All data were collected using standardized forms. As diagnostic accuracy, diagnostic odds ratio (DOR), areas under the curve (AUC), and summary receiver operating characteristic (SROC) were identified. The diagnostic performance was compared with histological examination results.

A random-effect model was used in this study. DOR represents the effectiveness of a diagnostic test. DOR is mathematically defined as (true positive/false positive)/(false negative/true negative). When DOR is greater than 1, higher values indicate better performance of the diagnostic method. A value of 1 means that the presence or absence of a disease cannot be determined and that the method cannot provide diagnostic information. To obtain an approximately normal distribution, we calculated the logarithm of each DOR and then calculated 95% confidence intervals [12]. SROC is a statistical technique used when performing a meta-analysis of studies that report both sensitivity and specificity. As the diagnostic ability of the test increases, the SROC curve shifts towards the upper-left corner of the ROC space, where both sensitivity and specificity are 1. AUC ranges from 0 to 1, with higher values indicating better diagnostic performance. We collected data on the number of patients, true-positive, true-negative, false-positive, and false-negative values in all included studies, and calculated AUCs and DORs from these values. The methodological quality of the included studies was evaluated using the Quality Assessment of Diagnostic Accuracy Study (QUADAS-2) tool.

### 2.4. Statistical Analysis and Outcome Measurements

R statistical software (R Foundation for Statistical Computing, Vienna, Austria) was used to conduct a meta-analysis of the studies. Homogeneity analyses were then performed using the Q statistic. Forest plots were drawn for the sensitivity, specificity, and negative predictive values, and for the SROC curves. A meta-regression analysis was performed to determine the potential influence of imaging tools on AI-based diagnostic accuracy for all premalignant lesions.

## 3. Results

This analysis included 14 studies [6,13,14,15,16,17,18,19,20,21,22,23,24,25]. Table 1 presents the assessment of bias. The characteristics of the studies are attached in Appendix A.

### 3.1. Diagnostic Accuracy of AI-Assisted Screening of Oral Mucosal Cancerous Lesions

Seven prospective and retrospective studies were included for discriminating oral cancerous lesions from normal mucosa. The diagnostic odds ratio (DOR) of AI-assisted screening was 121.6609 (95% confidence interval [CI], 29.5996; 500.0534, I^2^ = 93.5%) (Figure 2A).

The area under the summary receiver operating characteristic curve was 0.948, suggesting excellent diagnostic accuracy (Figure 3A).

The correlation between the sensitivity and the false-positive rate was 0.437, indicating the absence of heterogeneity. AI-assisted screening exhibited good sensitivity (0.9232 [0.8686; 0.9562]; I^2^ = 81.9%), specificity (0.9494 [0.7850; 0.9897], I^2^ = 98.3%), and negative predictive value (0.9405 [0.8947; 0.9671]. I^2^ = 83.6%) (Figure 4). The Begg’s funnel plot (Appendix A) shows that a source of bias was not evident in the included studies. The Egger’s test result (*p* > 0.05) also shows that the possibility of publication bias is low.

Subgroup analyses were performed to determine which image tool assisted by AI had higher discriminating power between oral cancer lesions and normal mucosa. This analysis showed that that there were no significant differences between the photographic image, autofluorescence, and OCT in AI based on the screening for oral cancer lesion (Table 2).

### 3.2. Diagnostic Accuracy of AI-Assisted Screening of Oral Mucosal Precancerous and Cancerous Lesions

Twelve prospective and retrospective studies were included for discriminating oral precancerous and cancerous lesions from normal mucosa. The diagnostic odds ratio (DOR) of AI-assisted screening was 63.0193 (95% confidence interval [CI], 40.3234; 98.4896, I^2^ = 88.2%) (Figure 2B). The area under the summary receiver operating characteristic curve was 0.943, suggesting excellent diagnostic accuracy (Figure 3B). The correlation between the sensitivity and the false-positive rate was 0.337, indicating the absence of heterogeneity. AI-assisted screening exhibited good sensitivity (0.9094 [0.8725; 0.9364]; I^2^ = 92.3%), specificity (0.8848 [0.8400; 0.9183], I^2^ = 93.8%), and negative predictive value (0.9169 [0.8815; 0.9424], I^2^ = 92.8%) (Figure 5).

The Egger’s test results of sensitivity (*p* = 0.02025) and negative predictive value (*p* < 0.001) also show that the possibility of publication bias is high. To compensate for the publication bias using statistical methods, trim-and-fill methods (trimfill) were applied to the outcomes. After implementation of trimfill, sensitivity dropped from 0.9094 [0.8725; 0.9364] to 0.8504 [0.7889; 0.8963] and NPV also dropped from 0.9169 [0.8815; 0.9424] to 0.7815 [0.6577; 0.8694]. These results could mean that the diagnostic power of AI-assisted screening of precancerous and cancerous lesions would be overestimated and clinicians would need to be careful when interpreting these outcomes.

Subgroup analyses were performed to determine which image tool assisted by AI had higher discriminating power of oral mucosal cancerous lesions including precancerous lesions. Subgroup analysis showed that OCT was more diagnostically accurate (324.3335 vs. 66.8107 and 27.6313) and more negatively predictive (0.9399 vs. 0.9311 and 0.8405) than photographic images and autofluorescence in AI based on the screening for oral precancerous and cancerous lesions from normal mucosa (Table 3). Meta-regression of AI diagnostic accuracy for oral precancerous and cancerous lesions on the basis of imaging tool revealed the significant correlations (*p* = 0.0050).

## 4. Discussion

Oral cancer is a malignant disease with high disease-related morbidity and mortality due to its advanced loco-regional status at diagnosis. Early detection of oral cancer is the most effective means to increase the survival rate and reduce morbidity, but a significant number of patients experience delays between noticing the first symptoms and receiving a diagnosis from a clinician [26]. In clinical practice, a conventional visual examination is not a strong predictor of oral cancer diagnosis, and a quantitatively validated diagnostic method is needed [27]. Radiographic imaging, such as magnetic resonance imaging and computed tomography, can help determine the size and extent of oral cancer before treatment, but these techniques are not sensitive enough to distinguish precancerous lesions. Accordingly, various adjunct clinical imaging techniques such as autofluorescence and OCT have been used [28].

AI has been introduced in various industries, including healthcare, to increase efficiency and reduce costs, and the performance of AI models is improving day by day [29]. For the past few years, the early detection of oral cancer using AI technology based on autofluorescence imaging, photographic imaging, and OCT imaging has been an important research area. In this study, diagnostic values including sensitivity and specificity data were comprehensively confirmed in various studies that performed AI analysis of images. The diagnostic sensitivity of oral cancer analyzed by AI was as high as 0.92, and the analysis including precancerous lesions was slightly lower than the diagnostic sensitivity for cancer, but this also exceeded 90%. In subgroup analysis, there was no statistically significant difference in the diagnostic rate according to each image tool. In particular, the sensitivity of OCT to all precancerous lesions was found to be very high at 0.94.

Autofluorescence images are created using the characteristic that autofluorescence naturally occurring from collagen, elastin, and other endogenous fluorophores such as nicotinamide adenine dinucleotide in mucosal tissues by blue light or ultraviolet light is expressed differently in cancerous lesions [30,31]. Although it has been used widely in the dental field for the purpose of screening abnormal lesions in the oral cavity, it has been reported that the accuracy is low, with a sensitivity of only 30–50% [32,33]. It has been noted that autofluorescence images have a low diagnostic rate when used in oral cancer screening. Most of the previous clinical studies on autofluorescence-obtained images used differences in spectral fluorescence signals between normal and diseased tissues. Recently, time-resolved autofluorescence measurements using the characteristics of different fluorescence lifetimes of endogenous fluorophores have been used to solve the problem of broadly overlapping spectra of fluorophores, improving image accuracy [34]. Using various AI algorithms for advanced autofluorescence images, the diagnostic sensitivity of precancerous and cancerous lesions was reported to be as high as 94% [15]. As confirmed in our study, AI diagnosis sensitivity using autofluorescence images was confirmed to be 85% in all precancerous lesions. It showed relatively low diagnostic accuracy when compared to other imaging tools in this study. However, autofluorescence imaging is of sufficient value as an adjunct diagnostic tool. Efforts are also being made to improve the diagnostic accuracy for oral cancer by using AI to analyze images obtained using other tools along with the autofluorescence image [19].

The photographic image is a fast and convenient method with high accessibility compared to other adjunct methods. However, there is a disadvantage in that the image quality varies greatly depending on the camera, lighting, and resolution used while obtaining the image. Unlike external skin lesions, the oral cavity is surrounded by a complex, three-dimensional structure including the lips, teeth, and buccal mucosa, which may decrease the image accuracy [6]. In a recent study introducing a smartphone-based device, it was reported that the problem of the image itself was solved through a probe that can easily access the inside of the mouth and increasing images pixel [35]. Image diagnosis using a smartphone is very accessible in the current era of billions of phone subscribers worldwide, and in particular, it is expected that accurate and efficient screening will be possible by diagnosing a vast number of these images with AI. According to our analysis, AI-aided diagnosis from photographic images was confirmed to have a diagnostic sensitivity of over 91% for precancerous and cancerous lesions.

OCT is a medical technology that images tissues using the difference in physical properties between the reference light path and the sample light path reflected after interaction in the tissue [13]. OCT is non-invasive and uses infrared light, unlike other radiology tests that use X-rays. It is also a good diagnostic method that allows real-time image verification. Since its introduction in 1991 [36], OCT has been developed to provide high-resolution images at a faster speed and has played an important role in the biomedical field. In an AI analysis study of OCT images published by Yang et al., it was reported that the sensitivity and specificity of oral cancer diagnosis was 98% or more [22]. In our study, OCT images were found to be the most accurate diagnostic test, with sensitivity of 94% in AI diagnosis compared to other image tools (sensitivity of autofluorescence and photographic images of 89% and 91%, respectively). Therefore, AI diagnosis using OCT images is considered to be of sufficient value as a screening method for oral lesions. Each image tool included in our study has its own pros and cons to be considered when using it in actual clinical practice. In addition, accessibility of equipment or systems that can be performed on patients in actual outpatient treatment will be an important factor.

Based on our results, AI analysis of images in cancer diagnosis is thought to be helpful in making fast decisions regarding further examination and treatment. The accuracy of discriminating between precancerous lesions and normal tissues showed a high sensitivity of over 90%, showing good accuracy as a screening method. Although the question of whether AI can replace experts still exists, it is expected that oral cancer diagnosis using AI will sufficiently improve mortality and morbidity due to disease in low- and middle-income countries with poor health care systems. Acquisition of large-scale image datasets to improve AI analysis accuracy will be a clinically important key.

Our study has several limitations. First, our results include data from multiple imaging tools analyzed at once. This created heterogeneity in the results. Therefore, the sensitivity of each imaging tool was checked separately. The study is meaningful as it is the first meta-analysis to judge the accuracy of AI-based image analysis. Second, even with the same imaging tool, differences in the quality of the devices used in each study and differences between techniques may affect the accuracy of diagnosis. The images used to train the AI algorithm may not fully represent the diversity of oral lesions. Third, there is a limit to the interpretation of the results due to the absolute lack of prospective studies between the conventional examination and AI imaging diagnosis. It is our task to study this in various clinical fields in order to prepare for a future in which AI-assisted healthcare will be successful

## 5. Conclusions

AI shows good diagnostic performance with high sensitivity for oral cancer. Through the development of image acquisition devices and the grafting of various AI algorithms, the diagnostic accuracy is expected to increase. As new studies in this field are published frequently, a comprehensive review of the clinical implications of AI in oral cancer will be necessary again in the future.

## Figures and Tables

**Figure 1 cancers-14-03499-f001:**
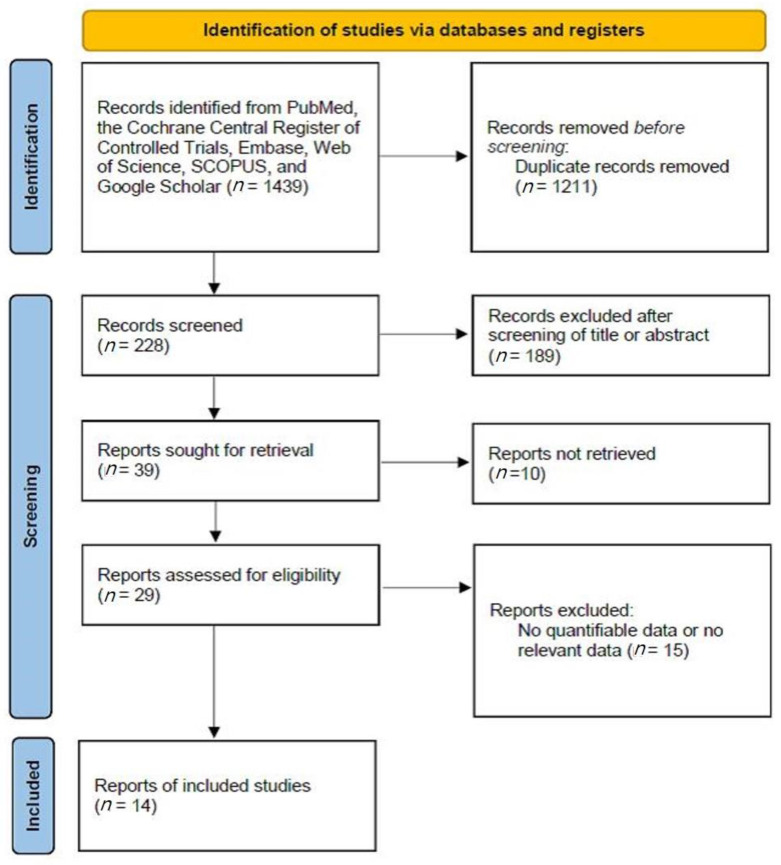
Summary of the search strategy.

**Figure 2 cancers-14-03499-f002:**
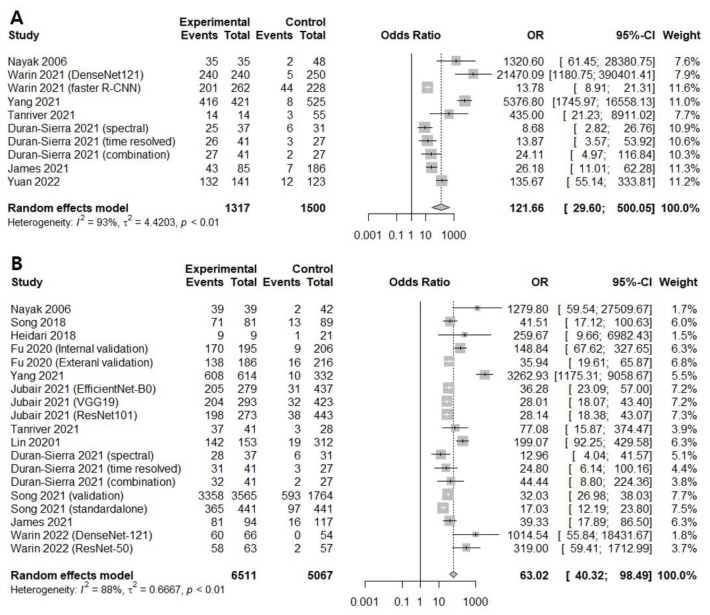
Forest plot of the diagnostic odds ratios for (**A**) screening only oral cancerous lesions [13,16,17,21,22,23,25] and (**B**) screening all premalignant mucosal lesions [13,14,15,16,17,18,19,20,21,23,24].

**Figure 3 cancers-14-03499-f003:**
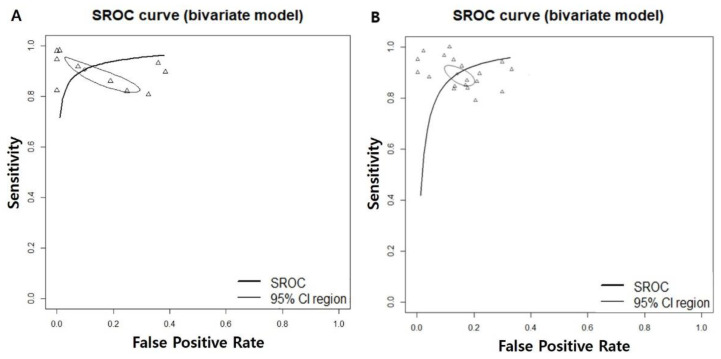
Area under the summary receiver operating characteristic for (**A**) screening only the oral cancerous lesions and (**B**) screening all premalignant mucosal lesions. SROC; summary receiver operating characteristic, CI; confidence interval.

**Figure 4 cancers-14-03499-f004:**
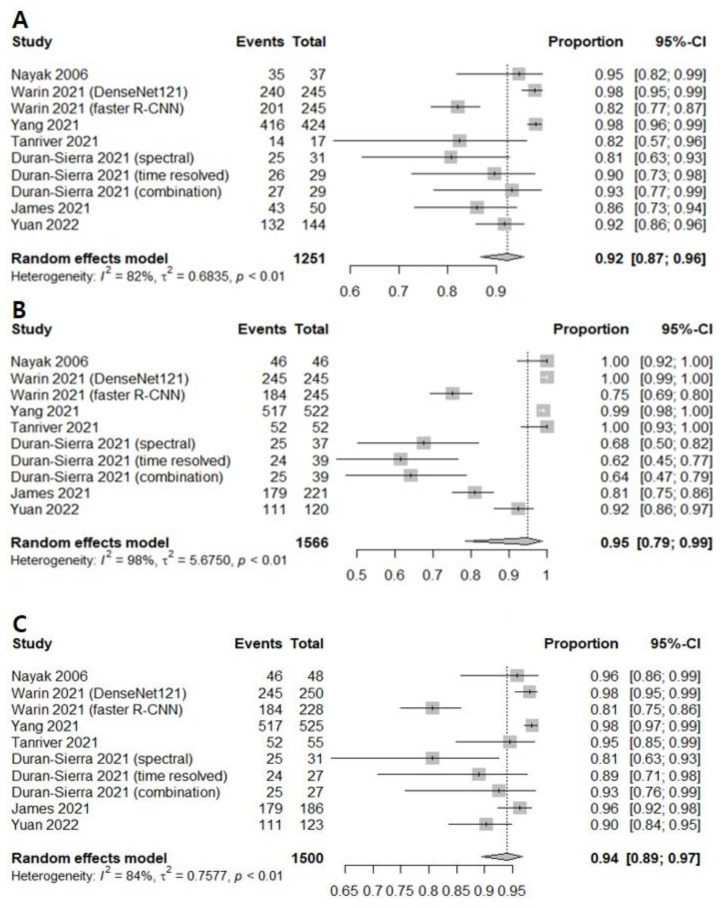
Forest plots of (**A**) sensitivity, (**B**) specificity, and (**C**) negative predictive values for screening oral cancerous lesions [13,16,17,21,22,23,25].

**Figure 5 cancers-14-03499-f005:**
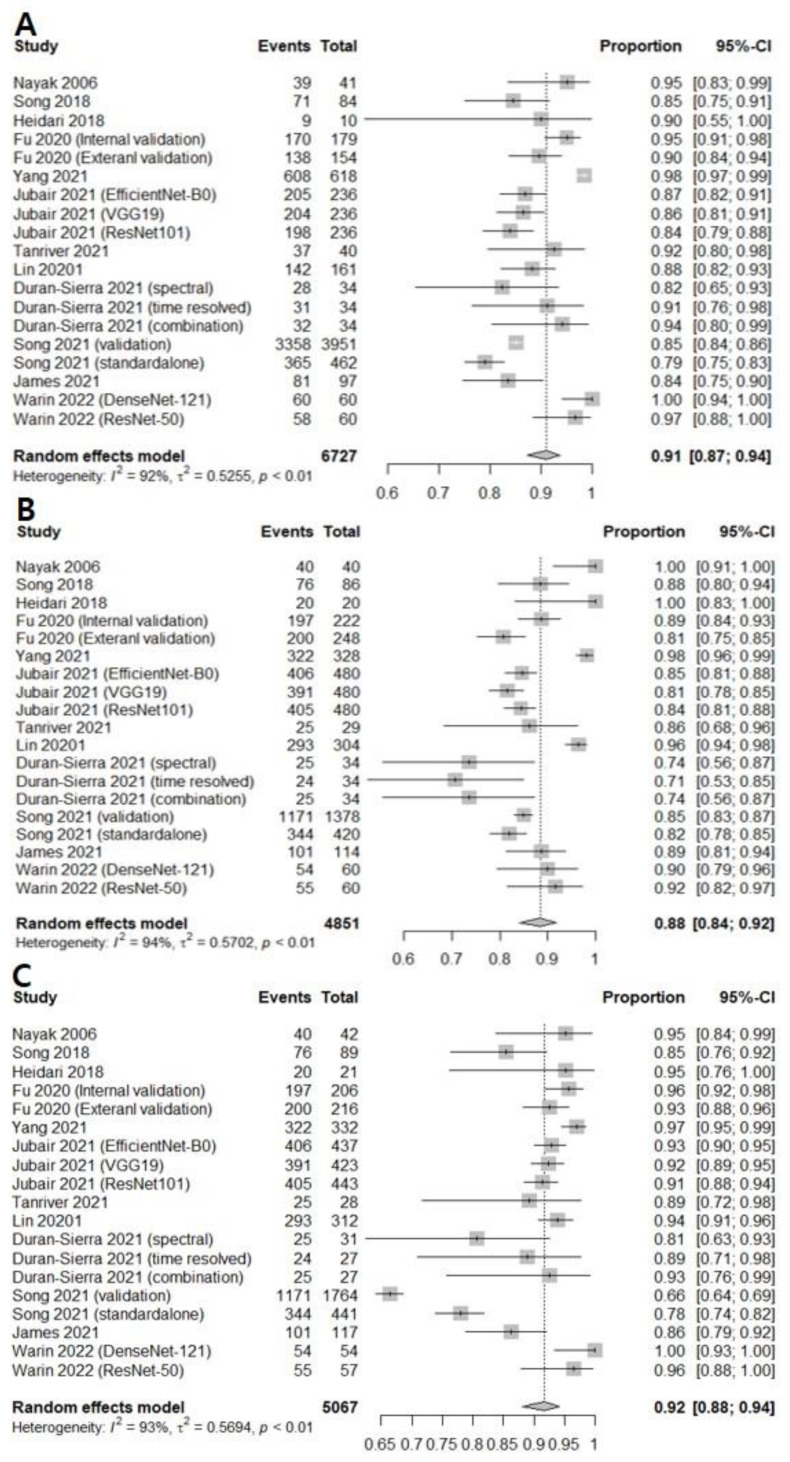
Forest plots of (**A**) sensitivity, (**B**) specificity, and (**C**) negative predictive values for screening all premalignant mucosal lesions [6,13,14,15,16,17,18,19,20,21,23,24].

**Table 1 cancers-14-03499-t001:** Methodological quality of all included studies.

Reference	Risk of Bias				Concerns about Application		
	Patient Selection	Index Test	Reference Standard	Flow and Timing	Patient Selection	Index Test	Reference Standard
Nayak 2006 [13]	Unclear	Low	Unclear	Unclear	Low	Low	Low
Heidari 2018 [14]	Low	Low	Low	Low	Low	Low	Low
Song 2018 [15]	Low	Low	Low	Low	Low	Low	Low
Fu 2020 [6]	high	Low	Low	Low	Low	Low	Low
Duran-Sierra 2021 [16]	Unclear	Low	Unclear	Unclear	Low	Low	Low
James 2021 [17]	Low	Low	Unclear	Low	Low	Low	Low
Jubair 2021 [18]	Unclear	Low	Low	Low	Low	Low	Low
Lin 2021 [19]	Unclear	Low	Unclear	Low	Low	Low	Low
Song 2021 [20]	Low	Low	Low	Low	Low	Low	Low
Tanriver 2021 [21]	Low	Low	Low	Low	Low	Low	Low
Warin 2021 [22]	Low	Low	Low	Low	Low	Low	Low
Yang 2021 [23]	Low	Low	Low	Low	Low	Low	Low
Warin 2022 [24]	Low	Low	Low	Unclear	Low	Low	Low
Yuan 2022 [25]	Low	Low	Low	Low	Low	Low	Low

**Table 2 cancers-14-03499-t002:** Subgroup analysis regarding image tool in discriminating oral cancerous lesions from normal mucosa.

Subgroup	Study (*n*)	DOR [95% CIs]	Sensitivity [95% CIs]	Specificity [95% CIs]	NPV [95% CIs]	AUC
	7	121.6609 [29.5996; 500.0534]; I^2^ = 93.5%	0.9232 [0.8686; 0.9562]; I^2^ = 81.9%	0.9494 [0.7850; 0.9897]; I^2^ = 98.3%	0.9405 [0.8947; 0.9671]; I^2^ = 83.6%	0.948
Image tool						
Autofluorescence	2	25.9083 [ 6.3059; 106.4464]; I^2^ = 68.0%	0.8972 [0.8262; 0.9413]; I^2^ = 63.5%	0.8213 [0.4430; 0.9637]; 94.0%	0.9041 [0.8263; 0.9492]; 23.9%	
Optical coherense tomography	3	261.9981 [14.7102; 4666.3521]; I^2^ = 96.3%	0.9419 [0.8544; 0.9781]; I^2^ = 84.4%	0.9461 [0.7931; 0.9877]; 94.6%	0.9625 [0.9106; 0.9848]; 81.9%	
Photographic image	2	431.6524 [ 4.0037; 46537.4743]; I^2^ = 93.0%	0.9149 [0.7475; 0.9750]; I^2^ = 87.4%	0.9983 [0.2906; 1.0000]; 94.9%	0.9381 [0.8109; 0.9816]; 87.5%	
		0.2332	0.5910	0.2907	0.2291	

DOR; diagnostic odds ratio, AUC; area under the curve, NPV; negative predictive value.

**Table 3 cancers-14-03499-t003:** Subgroup analysis regarding image tool in discriminating oral precancerous and cancerous lesions from normal mucosa.

Subgroup	Study (*n*)	DOR [95% CIs]	Sensitivity [95% CIs]	Specificity [95% CIs]	NPV [95% CIs]	AUC
	12	63.0193 [40.3234; 98.4896]; I^2^ = 88.2%	0.9094 [0.8725; 0.9364]; I^2^ = 92.3%	0.8848 [0.8400; 0.9183]; I^2^ = 93.8%	0.9169 [0.8815; 0.9424]; I^2^ = 92.8%	0.943
**Image tool**						
Autofluorescence	4	27.6313 [17.2272; 44.3186]; I^2^ = 69.3%	0.8562 [0.8002; 0.8985]; I^2^ = 69.6%	0.8356 [0.7591; 0.8913]; 86.8%	0.8405 [0.7487; 0.9031]; 91.1%	
Optical coherense tomography	3	324.3335 [10.2511; 10261.6006]; I^2^ = 95.6%	0.9424 [0.8000; 0.9853]; I^2^ = 88.3%	0.9653 [0.8737; 0.9911]; 79.8%	0.9399 [0.8565; 0.9762]; 75.7%	
Photographic image	5	66.8107 [38.0216; 117.3983]; I^2^ = 81.7%	0.9123 [0.8683; 0.9426]; I^2^ = 79.5%	0.8779 [0.8322; 0.9125]; 87.4%	0.9311 [0.9196; 0.9410]; 0.0%	
		0.0312	0.1120	0.0659	0.0073	

DOR; diagnostic odds ratio, AUC; area under the curve, NPV; negative predictive value.

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
