# Peer review of "Efficacy of Artificial Intelligence-Assisted Discrimination of Oral Cancerous Lesions from Normal Mucosa Based on the Oral Mucosal Image: A Systematic Review and Meta-Analysis"

_cancers, 2022, doi:10.3390/cancers14143499_

Round 1
Reviewer 1 Report
Suggest performing trial sequential analysis.
Suggest adding table for raw data (TP, TN, FP, FN) of the 7 and 12 studies.
Characteristics of enrolled studies e.g sample size, affiliation, study design, methods of AI used, age, gender, study groups ....of cohorts, year of publication.
Since there was heterogeneity that was not resolved by subgroup analysis, suggest applying meta-regression using study characteristics.
Reviewer 2 Report
The presented work is competently done and clearly described. Potentially, this review will be useful in the fast-developing area of AI-assisted medicine. The small number of reported cases still poses some questions on the reliability of some of the conclusions, as the authors dare to admit, but nevertheless the overall discussion and data analysis is coherent and timely.
Better image quality for figure 3 would be beneficial. And please check the number 3.5 % in the middle column in Table 2.
In conclusion, I would recommend publication of this review paper in its present form.
Author Response
[Reviewer #2]
The presented work is competently done and clearly described. Potentially, this review will be useful in the fast-developing area of AI-assisted medicine. The small number of reported cases still poses some questions on the reliability of some of the conclusions, as the authors dare to admit, but nevertheless the overall discussion and data analysis is coherent and timely.
Better image quality for figure 3 would be beneficial. And please check the number 3.5 % in the middle column in Table 2.
In conclusion, I would recommend publication of this review paper in its present form.
[Authors’ response to Reviewer #2]
Thank you for your comments.
Following your advice, we changed Figure 3 to a better resolution image.
Thanks for pointing out the typo in Table 2. We corrected it.
